# The Impact of the CYP2D6 and CYP1A2 Gene Polymorphisms on Response to Duloxetine in Patients with Major Depression

**DOI:** 10.3390/ijms241713459

**Published:** 2023-08-30

**Authors:** Julian Maciaszek, Tomasz Pawłowski, Tomasz Hadryś, Marta Machowska, Anna Wiela-Hojeńska, Błażej Misiak

**Affiliations:** 1Department of Psychiatry, Wroclaw Medical University, 50-367 Wroclaw, Poland; tomasz.pawlowski@umw.edu.pl (T.P.); tomasz.hadrys@umw.edu.wroc.pl (T.H.); blazej.misiak@umw.edu.pl (B.M.); 2Department of Clinical Pharmacology, Wroclaw Medical University, 50-367 Wroclaw, Poland; marta.machowska@umw.edu.pl (M.M.); anna.wiela-hojenska@umw.edu.pl (A.W.-H.)

**Keywords:** CYP2D6, CYP1A2, polymorphisms, duloxetine, tolerability, efficacy, major depression, anxiety symptoms, prediction

## Abstract

Depression is a global mental health concern, and personalized treatment approaches are needed to optimize its management. This study aimed to investigate the influence of the CYP2D6 and CYP1A2 gene polymorphisms on the efficacy of duloxetine in reducing depressive and anxiety symptoms. A sample of 100 outpatients with major depression, who initiated monotherapy with duloxetine, were followed up. Polymorphisms in the CYP2D6 and CYP1A2 genes were assessed. The severity of depressive and anxiety symptoms was recorded using standardized scales. Adverse drug reactions (ADRs) were analyzed. Statistical analyses, including linear regression, were conducted to examine the relationships between genetic polymorphisms, clinical variables, and treatment outcomes. Patients with higher values of the duloxetine metabolic index (DMI) for CYP2D6, indicating a faster metabolism, achieved a greater reduction in anxiety symptoms. The occurrence of ADRs was associated with a lower reduction in anxiety symptoms. However, no significant associations were found between studied gene polymorphisms and reduction in depressive symptoms. No significant effects of the DMI for CYP1A2 were found. Patients with a slower metabolism may experience less benefit from duloxetine therapy in terms of anxiety symptom reduction. Personalizing treatment based on the CYP2D6 and CYP1A2 gene polymorphisms can enhance the effectiveness of antidepressant therapy and improve patient outcomes.

## 1. Introduction

Major depression disorder (MDD) is a widely prevalent mental disorder, estimated to affect approximately 280 million people worldwide [1]. The World Health Organization (WHO) predicts that MDD will become the second leading cause of disability [2]. Severe depression can have fatal consequences, as about 60% of individuals who commit suicide suffer from depression [3]. Only 50–60% of patients respond to first-line treatments, and only 35–40% achieve symptom remission within an 8-week treatment period [4,5]. There is a considerable interindividual variability in terms of the efficacy and tolerability of treatments with antidepressants. This variability is attributable to a number of factors, including pharmacogenetic and pharmacodynamic mechanisms underlying the effects of antidepressants [6]. In this regard, personalized treatment offers the optimalization of the efficacy and overall costs of MDD treatment.

### 1.1. Duloxetine—The Mechanisms of Action

Serotonin and noradrenaline reuptake inhibitors (SNRIs), represented by duloxetine, serve as one of the most important classes of antidepressants. It has been shown that apart from their impact on serotoninergic and noradrenergic neurotransmission, SNRIs intensify dopaminergic transmission in the prefrontal cortex since dopamine is inactivated by norepinephrine reuptake (due to low amount of dopamine transporters in this brain region) [7]. The mechanism of action of SNRIs makes them suitable for the treatment of individuals with MDD manifesting in depressed mood, retardation, comorbid anxiety, somatic symptoms, and pain complaints. They also serve as an alternative for patients who do not respond to the treatment with selective serotonin reuptake inhibitors (SSRIs) [7,8]. In the case of duloxetine, other mechanisms may underlie its efficacy and originate from interactions with various neurotransmitter receptors, including those for acetylcholine, noradrenaline, dopamine, and histamine [8]. Both short-term and long-term clinical trials have confirmed the efficacy of duloxetine in the treatment of severe depressive episodes and have determined its effectiveness to be comparable to other selective serotonin inhibitors, such as fluoxetine, paroxetine, citalopram, and sertraline [9,10].

### 1.2. Pharmacogenetics of Duloxetine

Duloxetine is mainly metabolized by mitochondrial isoforms of cytochrome P450 izoenzymes: CYP2D6 and CYP1A2 [11]. Based on the enzymatic activity of various CYP2D6 polymorphisms, five phenotypes have been identified: ultrarapid metabolizers, rapid metabolizers, normal metabolizers, intermediate metabolizers, and poor metabolizers [12,13]. Poor metabolizers present significantly higher concentrations of antidepressant drugs compared to other groups. However, they are more prone to treatment side effects, while ultrarapid metabolizers typically show subtherapeutic drug concentrations, leading to reduced efficacy of antidepressant treatment [14]. So far, 172 variants of the CYP2D6 gene have been identified [15]. The CYP1A2 is involved in the metabolism of several clinically important drugs, including antidepressants. To date, 21 various CYP1A2 alleles and corresponding metabolic subtypes have been described [16].

So far, only one but very promising clinical study from 2020 evaluating the impact of duloxetine metabolism on clinical aspects has been published by Zastrozhin et al. [17]. In this study, the effects of the CYP2D6 gene polymorphic marker (1846 G > A, rs3892097) were found, i.e., patients carrying the A allele had a lower level of drug equilibrium concentration than those with the G allele. Patients with the GA genotype, indicative of poor duloxetine metabolism, showed higher levels of depressive symptoms and experienced more adverse effects compared to patients with the GG genotype and standard metabolism. Researchers investigated the effects of polymorphisms on depressive symptoms, but not anxiety symptoms. To bridge this gap, we designed the study that also focused on anxiety symptoms, frequently associated with MDD. Specifically, the present study aimed to assess the influence of the CYP2D6 and CYP1A2 gene polymorphisms on the effectiveness of duloxetine in reducing depressive and anxiety symptoms among patients with MDD.

## 2. Results

### 2.1. General Characteristics of the Sample

The general characteristics of the sample are presented in Table 1.

The mean age of individuals participating in the study was 41.6 ± 18.4 years. The participants were predominantly females (57.0%) and individuals with a higher level of education (65.0%). The mean score for the baseline HAM-D was 21.6 ± 9.3, and the mean reduction in the HAM-D was 52.3 ± 40.9%. The mean score for the baseline HAM-A was 17.3 ± 7.5, and the mean reduction in the HAM-A was 51.1 ± 37.4%. The UKU-SERS score was 5.8 ± 7.6 points. Four patients discontinued the therapy due to severe adverse drug reactions, including vomiting, sedation, insomnia, worsening of depression, increased anxiety, and emotional blunting. Figure 1 shows a flow diagram of study participants.

A total of 48 patients reached their final dose of duloxetine after 1 month of therapy. Among them, four individuals maintained the dosage of 30 mg, while the remaining forty-four participants adhered to the treatment protocol with a daily dose of 60 mg. After two months, 25 patients continued the treatment and received 90 mg of duloxetine on a daily basis. Subsequently, 27 patients completed the treatment taking 120 mg of duloxetine daily.

Appendix A shows the distribution of gene polymorphism variants at the CYP2D6 gene in two alleles (4*1846G→A and 3*2550delA) and the CYP1A2 in three alleles (1F*-163C→A, 3*1545T→C, and 1D*-24124delT), together with the corresponding DMS ranging from 1 to 5. The DMI of the studied sample (n = 100) for the CYP2D6 was found to be 2.8 ± 0.3, while for the CYP1A2 it was 3.1 ± 0.2.

### 2.2. Stepwise Linear Regression Analyses

Linear regression analysis revealed significant positive associations between the CYP2D6 DMI and the percentage of change in the HAM-A score across all models (Table 2).

Also, there was a significant negative association between the UKU-SERS score and the percentage of change in the HAM-A score. No significant associations between the CYPA1A2 DMI and the percentage of change in the HAM-D score were found across all models (Table 3). The association between the UKU-SERS score and the percentage of change in the HAM-D score was significant and negative only in the second model. No significant collinearity was observed.

## 3. Materials Methods

The current study included a cohort of 100 Caucasian individuals who were not taking any medications and presented with an active episode of MDD. These patients were initiated on monotherapy with duloxetine, starting with low doses and gradually increasing according to the study protocol. This study was a part of a larger research project exploring the efficacy and tolerability of duloxetine in relation to genetic polymorphisms. The participants did not receive any additional psychiatric medications. The study took place at an outpatient clinic located in the Department of Psychiatry (Wroclaw Medical University, Wroclaw, Poland). The inclusion criteria required participants to be between 18 and 85 years of age, have no ongoing psychiatric treatment, and meet the diagnostic criteria for MDD according to the ICD-10 criteria, with a duration of at least 14 days. The exclusion criteria were a lifetime diagnosis of schizophrenia, bipolar disorder, dementia, substance use disorders (except of nicotine dependence), severe or unstable somatic conditions, and noncompliance with duloxetine monotherapy. The patients participated in regular psychiatric consultations every 4 weeks, during which the dosage was adjusted according to the protocol described in detail in the previous publication [18]. The study was approved by the Ethics Committee at Wroclaw Medical University (approval number: 606/2017). Demographic data, such as age, sex, the level of education, and marital status, along with the age of MDD onset, presence of chronic somatic conditions, and vocational status, were collected as additional measures. The severity of depressive symptoms was assessed using the Hamilton Depression Rating Scale (HAM-D), a 21-item questionnaire that focuses on somatic symptoms and anxiety to measure the severity of depressive symptoms [19]. Similarly, the severity of anxiety symptoms was evaluated utilizing the Hamilton Anxiety Rating Scale (HAM-A), which comprises 14 items [20]. Adverse drug reactions (ADRs) occurring within the preceding 3 days were recorded through the use of the Udvalg for Kliniske Undersogelser Side Effect Rating Scale (UKU SERS), which consists of 48 items [21]. Each item was scored on a 4-point scale, ranging from “no, not at all” to “much more than usual”. Patient feedback was used to document ADRs for each parameter listed in the UKU SERS scale. These items were then categorized into four sub-groups, namely psychic, neurological, autonomic, and other side effects. After each clinical assessment, it was evaluated as to whether specific ADRs could be attributed to duloxetine treatment. The total UKU-SERS score was calculated by summing the points for ADRs associated with duloxetine treatment.

### 3.1. Genetic Polymorphisms Analysis

Alleles selected for further research were the most common functionally important alleles in the Caucasian population, selected on the basis of global databases, including PharmGKB [22] and CPIC guidelines [23] according to the state of knowledge current for 2017, when the study was planned. We decided to narrow the scope of our study to the two polymorphisms of the CYP2D6 gene and three in the CYP1A2 gene. For Caucasians, the CYP2D6 allele *3 has a mean frequency of 1.6% and the allele *4 of 18.5%, whereas the CYP1A2 allele 1F has a mean frequency of 33% and the allele *3 of 1%. In turn, the frequency of the CYP1A2 allele 1D varies between 3.4 and 11% [24,25,26].

Venous blood was collected using the Vacutainer system (Becton Dickinson) into the K2-EDTA anticoagulant tubes. The analysis of the CYP2D6 and CYP1A2 gene polymorphisms was performed by isolating genetic material using a column-based kit, followed by the polymerase chain reaction—restriction fragment length polymorphism (PCR-RFLP) with agarose gel electrophoresis using the GCMHU-502 electrophoresis kit (C.B.S. Scientific). Within the CYP2D6 gene, polymorphisms in two alleles (4*1846G→A and 3*2550delA) were analyzed. For the CYP1A2 gene, polymorphisms of three alleles, including 1F*-163C→A, 3*1545T→C, and 1D*-24124delT were analyzed (see Appendix A for details about genotyping procedures).

The author’s duloxetine metabolic score (DMS) was used to describe the effect of individual gene polymorphisms on duloxetine metabolism. The DMS ranged from 1 to 5, representing a poor metabolizer (DMS = 1), intermediate metabolizer (DMS = 2), normal metabolizer (DMS = 3), rapid metabolizer (DMS = 4), and ultrarapid metabolizer (DMS = 5) [27]. The author’s duloxetine metabolic index (DMI) describing the cumulative effect of the studied polymorphisms separately for the CYP2D6 and CYP1A2 on the metabolism of duloxetine was then calculated from the DMS as the arithmetic mean of the corresponding polymorphisms. The CYP2D6 DMI was calculated as the arithmetic mean of the 2 DMS values for CYP2D6, and the DMI for CYP1A2 was calculated as the arithmetic mean of the 3 corresponding DMS values for CYP1A2 [28].

### 3.2. Statistical Analysis

Before performing the analyses, data were inspected for outliers using the boxplots and Q–Q plots. Two stepwise linear regression analyses were carried out. The first one included the percentage of reduction in the HAM-A score, while the second one included the percentage of reduction in the HAM-D score as dependent variables. The stepwise procedure included three blocks of independent variables: (1) the DMI for CYP2D6 and CYP1A2; (2) potential covariates related to duloxetine treatment (treatment duration, duloxetine dosage, and treatment tolerance expressed by the UKU-SERS total score); and (3) sociodemographic and clinical characteristics (age, sex, marital status, the level of education, and the presence of chronic somatic diseases). Collinearity was assessed by calculating the variance inflation factor (VIF). Significant collinearity was defined as VIF > 4. The statistical analysis was conducted using the SPSS software, version 28. The level of significance was set at *p* < 0.05.

## 4. Discussion

The aim of this study was to assess the influence of the CYP2D6 and CYP1A2 gene polymorphisms on the efficacy of duloxetine treatment in reducing depressive and anxiety symptoms among patients with MDD. In terms of the reduction in anxiety symptoms, we found a significant impact of two factors: the CYP2D6 DMI and UKU-SERS. Patients with higher values of the CYP2D6 DMI, indicating faster metabolism of duloxetine, achieved a greater reduction in anxiety symptoms as a result of the treatment. Additionally, a higher occurrence of adverse events was associated with a lower reduction in anxiety symptoms. We found no significant association between the CYP2D6 DMI and the reduction in depressive symptoms. The CYP1A2 DMI did not have a significant impact on the reduction in anxiety and depressive symptoms.

Our findings indicate a significant role of the CYP2D6 gene polymorphisms in predicting the reduction in anxiety disorders, interestingly without a clear influence on the reduction in depressive symptoms during duloxetine therapy among patients with MDD. Patients with a slow metabolism of duloxetine in relation to the CYP2D6 experienced a lower reduction in anxiety symptoms. Our findings also highlight the significant impact of the occurrence of adverse events on the degree of reduction in anxiety symptoms. Considering data on the increased likelihood of adverse events in patients with a slow metabolism of duloxetine (the GA and GG genotypes within the CYP2D6), who have relatively higher drug concentrations in their blood at similar doses [17], it is likely that the potentially more frequent occurrence of adverse events in this group could lead to a greater severity of anxiety reactions. Consequently, this could result in a lower reduction in anxiety disorders among patients with a slow duloxetine metabolism. In the case of depressive symptoms, a higher intensity of adverse events was also associated with a lower reduction in depressive symptoms, but no significant association with the CYP2D6 DMI and CYP1A2 DMI was observed, and the effect appeared to be not significant after expanding the regression model to include sociodemographic variables. Our findings partly correspond to those obtained by Zastrozhin et al. [17], with the difference that in the cited study a slow metabolism of the CYP2D6 resulted in a smaller reduction in depressive symptoms on the HAM-D scale, whereas in our study no significant association was found with respect to depressive symptoms on the HAM-D. However, a significant association was observed between a slower metabolism and a smaller reduction in symptoms of anxiety disorders on the HAM-A scale, which has not been assessed in a similar study so far. These discrepancies may originate from variations in allele distribution and methodological differences. In the study by Zastrozhin et al. [17], which included a similar group of patients (n = 118), no phenotypes of ultraslow metabolism associated with the AA alleles were identified, while in our study six patients had this metabolic genotype. These differences may be due to variations in the populations studied and would certainly be minimized with the inclusion of larger patient groups. Regarding metabolic differences, it is important to note that the study by Zastrozhin et al. [17] also included patients with a history of alcohol dependence, which was an exclusion criterion in our study. Alcohol dependence may be associated with dysfunction of the reward system in the brain, which can affect the efficacy of antidepressant treatment and may partially explain the observed differences between the compared groups [29]. It is further important to note the potentially impaired liver function in this group of patients, which can affect efficacy and tolerability. In the study by Zastrozhin et al. [17], a slow metabolism of the CYP2D6 was associated with changes in duloxetine concentration in the blood and higher levels of side effects during duloxetine therapy, which significantly affected the reduction in anxiety and depressive symptoms in our study. Also, in our study, the lack of significant effects of the CYP1A2 metabolism on clinical variables can be explained by its smaller contribution to duloxetine metabolism, estimated at approximately 20–30% compared to 70–80% for the CYP2D6 [30].

Our findings might also be interpreted in light of evidence from studies of venlafaxine. Venlafaxine is primarily metabolized by the CYP2D6, leading to the production of pharmacologically active metabolite—O-desmethylvenlafaxine (ODV) and, to a lesser extent by the CYP3A4, contributing to the release of N-desmethylvenlafaxine (NDV). In the study by Shams et al. [31], it was found that the ratio of venlafaxine metabolite concentrations significantly influenced the occurrence of adverse effects: the ODV/NDV ratios below 0.3 were associated with more side effects. However, no significant impact of the CYP2D6 or CYP3A4 gene polymorphisms on the antidepressant treatment efficacy was observed. According to Sangkhul et al. [32], individuals who are poor CYP2D6 metabolizers have increased levels of venlafaxine and decreased levels of ODV, which appears to be associated with a higher risk of side effects and a reduced response to therapy. The impact of the genetic variation in the CYP2D6 gene regarding venlafaxine metabolism is believed to be particularly pronounced in older patients [33,34]. In the study by Tarnau et al. [35], conducted in a group of 206 patients, no significant association between the CYP2D6 metabolism variants and changes in the HAM-A scale was observed. Similarly, in the posthoc analysis of a randomized clinical trial, no significant influence of rapid or slow CYP2D6 metabolism on depressive treatment efficacy was observed [36].

It is also important to consider the potential phenomenon of cytochrome P450 induction. Notably, CYP1A2 has been documented to demonstrate inducibility at both the mRNA and protein levels in response to a spectrum of chemical agents, including omeprazole, lansoprazole, 2,3,7,8-tetrachlorodibenzo-*p*-dioxin (TCDD), 3-methylcholanthrene, and rifampicin [37]. However, it should be noted that none of the subjects enrolled in this study were administered these specific drugs. Additionally, it is noteworthy that some reports suggest an augmentation of the CYP2D6 activity, typically considered noninducible, following exposure to common valerian (in a linear dose-dependent manner) and Ginkgo biloba (with the highest concentration) extracts [38]. It is essential to highlight that herbal dietary supplements were not permitted within the study cohort. Furthermore, in the context of drug metabolism, it is relevant to acknowledge the influence of cigarette smoking, which can expedite the metabolism of certain drugs, particularly those predominantly metabolized by CYP1A2. Notwithstanding, as demonstrated in the study by Hukkanen et al. [39], it should be clarified that nicotine itself does not contribute to the induction of CYP1A2. In sum, the phenomenon of metabolic induction and phenoconversion likely had a negligible impact on the results of our study.

### Limitations

The present study has certain limitations. First, considering the requirements associated with analyses related to genetic polymorphisms, our sample size was relatively small. Additionally, the absence of a double-blind randomized clinical trial design can be considered another limitation. Furthermore, the measures of treatment adherence were not included. Moreover, certain characteristics that could be associated with treatment efficacy, such as personality traits, treatment adherence, and the level of illness insight, were not recorded, which could have influenced the observed associations. It is worth noting that individuals with first-episode depression and those with recurrent depression may show differences in their responses to antidepressants. However, due to the small sample size, this aspect was not addressed in the present study. Also, the limited duration of the study does not allow to provide insights into long-term predictors of response to duloxetine treatment. Finally, the reported findings should be interpreted with caution as the Food and Drug Administration has recognized that only concomitant administration of duloxetine with the potent CYP1A2 inhibitor (such as fluvoxamine) to CYP2D6 poor metabolizers results in the altered pharmacokinetics of duloxetine [40]. In our study, we did not take into account the CYP2D6 and CYP1A2 variants sporadically occurring in the Polish population, which should be perceived as a limitation. The limitation of this study also stems from its methodological necessity to examine patients exclusively in a monotherapy context, whereas in clinical practice, monotherapy represents a less common treatment modality for depression.

## 5. Conclusions

In summary, the results of our study indicate a significant influence of CYP2D6 metabolism on the reduction in anxiety symptoms, without a distinct impact on the reduction in depressive symptoms. This can be explained by the mechanism of drug metabolism affecting the occurrence of adverse effects and, subsequently, triggering anxiety symptoms. The obtained results can be applied in the personalization of duloxetine treatment, i.e., poor and intermediate duloxetine metabolism within the CYP2D6 and concomitant high severity of anxiety symptoms could potentially be considered as a relative contraindication for duloxetine use, while rapid and ultrarapid duloxetine metabolism within CYP2D6 may be potentially associated with a good response in patients with MDD and high severity of comorbid anxiety disorders. However, further research is needed to confirm the predictive influence of the CYP2D6 variability on the efficacy of duloxetine treatment in reducing symptoms associated with MDD.

## Figures and Tables

**Figure 1 ijms-24-13459-f001:**
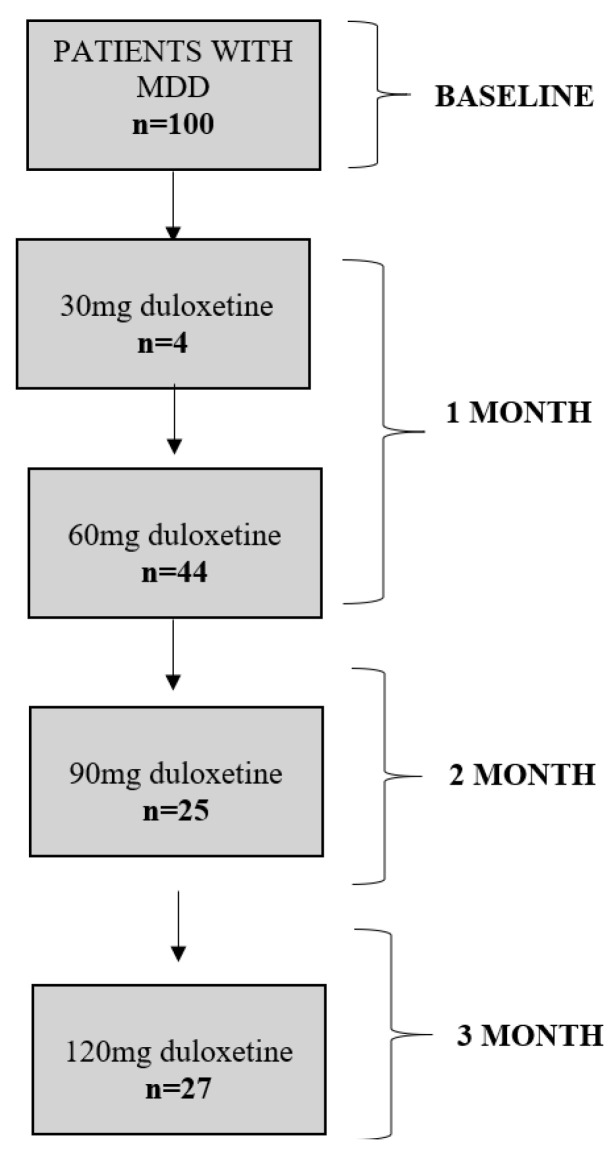
Flow diagram of study participants.

**Table 1 ijms-24-13459-t001:** General characteristics of the sample.

Variable	Mean ± SD or n (%)
Age	41.6 ± 18.4
Sex (female)	57 (57.0%)
Marital status (married)	37 (37.0%)
Education (higher education)	65 (65.0%)
Somatic conditions	30 (30.0%)
Diabetes drugs	5 (5.0%)
Hypotensive drugs	12 (12.0%)
Anti-asthmatic drugs	6 (6.0%)
Chronic obstructive pulmonary disease drugs	4 (4.0%)
Coronary artery disease drugs	3 (3.0%)
Rheumatoid arthritis drugs	2 (2.0%)
Drugs for chronic migraines	8 (8.0%)
Drugs for osteoarthritis	5 (5.0%)
Drugs for chronic pain	7 (7.0%)
Other drugs	5 (5.0%)
Cigarette smoking	27 (27.0%)
UKU SERS	5.8 ± 7.6
HAM-D (baseline score)	21.6 ± 9.3
HAM-D (reduction, %)	52.3 ± 40.9
HAM-A (baseline score)	17.3 ± 7.5
HAM-A (reduction, %)	51.1 ± 37.4

Data expressed as n (%) or mean (SD).

**Table 2 ijms-24-13459-t002:** Linear stepwise regression. Dependent variable: HAM-A reduction (%).

Model (R Square)	Variable	Beta	t	*p*	VIF
DMI (R^2^ = 0.052)	CYP2D6 DMI	0.216	2.181	0.032	1.005
	CYP1A2 DMI	0.086	0.867	0.388	1.005
DMI and potential covariates related to duloxetine treatment (R^2^ = 0.119)	CYP2D6 DMI	0.208	2.108	0.038	1.040
	CYP1A2 DMI	0.077	0.786	0.434	1.035
	UKU SERS	−0.222	−2.108	0.038	1.185
	Dose	−0.203	−1.779	0.078	1.393
	Time	0.060	0.504	0.615	1.533
DMI, potential covariates related to duloxetine treatment, sociodemographic and clinical characteristics (R^2^ = 0.150)	CYP2D6 DMI	0.205	2.041	0.044	1.052
	CYP1A2 DMI	0.100	0.938	0.351	1.185
	UKU SERS	−0.236	−2.163	0.033	1.244
	Dose	−0.207	−1.647	0.103	1.647
	Time	0.039	0.306	0.760	1.699
	Age	−0.091	−0.661	0.510	1.973
	Sex	0.032	0.305	0.761	1.143
	Marital Status	−0.046	−0.372	0.711	1.624
	Somatic Disease	0.117	1.098	0.275	1.195
	Education	−0.132	−1.227	0.223	1.219

Abbreviation: DMI—duloxetine metabolic index; UKU SERS—Udvalg for Kliniske Undersogelser Side Effect Rating Scale.

**Table 3 ijms-24-13459-t003:** Linear stepwise regression. Dependent variable: HAM-D reduction (%).

Model (R Square)	Variable	Beta	t	*p*	VIF
DMI (R^2^ = (0.019)	CYP2D6 DMI	0.129	1.281	0.203	1.005
	CYP1A2 DMI	0.054	0.538	0.592	1.005
DMI and potential covariates related to duloxetine treatment (R^2^ = 0.070)	CYP2D6 DMI	0.107	1.051	0.296	1.040
	CYP1A2 DMI	0.066	0.656	0.514	1.035
	UKU SERS	−0.231	−2.132	0.036	1.185
	Dose	−0.146	−1.241	0.218	1.393
	Time	0.160	1.297	0.198	1.533
DMI, potential covariates related to duloxetine treatment, sociodemographic and clinical characteristics (R^2^ = 0.080)	CYP2D6 DMI	0.104	0.997	0.322	1.052
	CYP1A2 DMI	0.060	0.542	0.589	1.185
	UKU SERS	−0.216	−1.909	0.060	1.244
	Dose	−0.155	−1.189	0.238	1.647
	Time	0.156	1.180	0.241	1.699
	Age	−0.048	−0.337	0.737	1.973
	Sex	0.007	0.064	0.949	1.142
	Marital Status	−0.008	−0.059	0.953	1.624
	Somatic Disease	−0.032	−0.290	0.772	1.195
	Education	−0.128	−1.143	0.256	1.219

Abbreviation: DMI—duloxetine metabolic index; UKU SERS—Udvalg for Kliniske Undersogelser Side Effect Rating Scale.

## Data Availability

The data presented in this study are available on request from the corresponding author. The data are not publicly available due to protection of the results of genetic assays.

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
