# Peer review of "The Impact of the CYP2D6 and CYP1A2 Gene Polymorphisms on Response to Duloxetine in Patients with Major Depression"

_ijms, 2023, doi:10.3390/ijms241713459_

Round 1
Reviewer 1 Report
The manuscript “The impact of the CYP2D6 and CYP1A2 Gene Polymorphisms on Response to Duloxetine in Patients with Major Depression” addresses how some SNPs of these two genes influence depressive and anxiety symptoms of patients with major depression receiving duloxetine treatment in monotherapy. Although duloxetine is truly metabolized by CYP2D6 and CYP1A2, FDA only recognizes a drug label that includes that concomitant administration of this drug with a potent CYP1A2 inhibitor (such as fluvoxamine) to CYP2D6 poor metabolizers results in an altered pharmacokinetics of duloxetine. Authors don’t accurately analyze previous literature and pharmacogenetics evidence such as CPIC guidelines or PharmGKB. In methods there is a lack of protocols for genotyping CYP2D6 and CYP1A2 (primers, PCR conditions, restriction endonucleases). Ethnicity of patients and prevalence of this specific SNPs wihtin their ethnicity are not shown. Authors should not generalize when talking about CYP2D6 in general since CYP2D6 is highly polymorphic and only two variants are analysed. Moreover, monotherapy is not generally used for patients with major depression, and this fact should be explained as a limitation of the study. As a minor aspect, there is the lack of a reference in line 101. The format of references should be revised.
Author Response
The manuscript “The impact of the CYP2D6 and CYP1A2 Gene Polymorphisms on Response to Duloxetine in Patients with Major Depression” addresses how some SNPs of these two genes influence depressive and anxiety symptoms of patients with major depression receiving duloxetine treatment in monotherapy. Although duloxetine is truly metabolized by CYP2D6 and CYP1A2, FDA only recognizes a drug label that includes that concomitant administration of this drug with a potent CYP1A2 inhibitor (such as fluvoxamine) to CYP2D6 poor metabolizers results in an altered pharmacokinetics of duloxetine.
Thank you for this crucial remark.
We have added the following sentence in discussion subsection:
“The reported findings should be interpreted with caution as the Food and Drug Administration has recognized that only concomitant administration of duloxetine with the potent CYP1A2 inhibitor (such as fluvoxamine) to CYP2D6 poor metabolizers results in an altered pharmacokinetics of duloxetine [40]. “
Authors don’t accurately analyze previous literature and pharmacogenetics evidence such as CPIC guidelines or PharmGKB.
Thank you for this valuable remark. We have added following information in the Methods section:
“Alleles selected for further research were the most common functionally important alleles in the Caucasian population, selected on the basis of global databases, including PharmGKB [22] and CPIC guidelines [23] according to the state of knowledge current for 2017, when the study was planned. We decided to narrow the scope of our study to the two polymorphisms of the CYP2D6 gene and three in the CYP1A2 gene.”
In methods there is a lack of protocols for genotyping CYP2D6 and CYP1A2 (primers, PCR conditions, restriction endonucleases).
Thank you for this valuable suggestion. We have provided a new Table S1 with detailed protocols:
Table S1. Detailed protocols for genotyping CYP2D6 and CYP1A2
|
Polymorphism |
Primers |
Preliminary denaturation |
Amplification |
Elongation |
Restriction |
|
CYP1A2 *1D |
Cyp1A21DF 5′-TGA GCC ATG ATT GTG GCA TA-3 ′ CYP1A21DR 5′-AGG AGT CTT TAA DATA GGA CCC AG-3 ′
|
5 min, 94 ° C |
35 cycles 30 s, 94 ° C 10 s, gradient 49 ° C 30 s, 72 ° C |
5 min, 72 ° C |
Ndei Eurx* |
|
CYP1A2 *3 |
Cyp1A23F 5'-AGC CCT TGA GTG AGA AGA TG-3 ' Cyp1A23r 5'- GGT CTT GCT CTG TCA CTC A-3 '
|
5 min, 94 ° C |
35 cycles 30 s, 94 ° C 10 s, gradient 58 ° C 30 s, 72 ° C |
5 min, 72 ° C |
Mluci New England Biolabs |
|
CYP1A2 *1F |
CYP1A2FF 5′-CAC CCT GCC AAT CTC AAG CAC-3 ′ CYP1A21FR 5′-AGA AGC TCT GTG GCC GAG AAG G-3 ′
|
3 min, 94 ° C |
30 cycles 10 s, 98 ° C 30 s, 55 ° C 1 min, 72 ° C
|
5 min, 72 ° C |
APAI New England Biolabs |
|
CYP2D6 *4 |
Primer C: 5'-GCC TTC GCC AAC CAC TCC G-3 ' Primer D: 5'- AAA TCC TGC TCT TCC GAG GC-3 ’
|
1 min, 94 ° C |
35 cycles 30 s, 94 ° C 30 s, 59 ° C 30 s, 72 ° C
|
5 min, 72 ° C |
Wales, New England Biolabs |
|
CYP2D6 *3 |
Primer E: 5'-gat gag ctg cta act gag ccc-3 ' Primer F: 5'-CCG AGA GCA TAC TCG GGA C-5 '
|
1 min, 94 ° C |
35 cycles 30 s, 94 ° C 30 s, 59 ° C 30 s, 72 ° C
|
5 min, 72 ° C |
Hpaii, Eurx |
*Due to the difficulty of reading restrictive stripes on agarose gel the TAQman SNP set was also used in identifying the presence of the allel *1D Genotyping Assay C__60142977_10, Thermo Fischer Scientific
Ethnicity of patients and prevalence of this specific SNPs wihtin their ethnicity are not shown.
Thank you for your vigilance, we have added the necessary information about ethnicity:
“For Caucasians, the CYP2D6 allele *3 has the mean frequency of 1.6% and the allele *4 of 18.5% whereas the CYP1A2 allele 1F has the mean frequency of 33% and the allele *3 of 1%. In turn, the frequency of the CYP1A2 allele 1D varies between 3.4 and 11% [24-26].”
Authors should not generalize when talking about CYP2D6 in general since CYP2D6 is highly polymorphic and only two variants are analysed.
Thank you for this crucial remark. We have highlighted this point as a limitation of our study:
“In the described study, we did not take into account the CYP2D6 and CYP1A2 variants sporadically occurring in the Polish population, which should be perceived as a limitation.”
Moreover, monotherapy is not generally used for patients with major depression, and this fact should be explained as a limitation of the study.
Thank you for this crucial remark. We have included this point as a limitation of our study:
“The limitation of this study stems also from its methodological necessity to examine patients exclusively in a monotherapy context, whereas in clinical practice, monotherapy represents a less common treatment modality for depression.”
As a minor aspect, there is the lack of a reference in line 101. The format of references should be revised.
Thank you for your vigilance . We have added necessary references and revised the format of references.
Reviewer 2 Report
- In this manuscript, the authors provide an overview of their study associating duloxetine response with CYP2D6 and CYP1A2 metabolizing capacity (using calculated scores reflecting predicted enzyme activity), side effects, and a number of additional clinical factors such as somatic conditions and sex. They determined that higher CYP2D6 metabolism and less side effect burden were associated with better response.
- This is an interesting and timely paper given that pharmacogenetic panel tests being sold to guide psychotropic treatment sometimes include CYP1A2 and CYP2D6 when formulating recommendations for duloxetine use, based on limited evidence
- However, there are several issues that should be addressed before publication
- The data on number of CYP2D6 and CYP1A2 variants is outdated: https://www.pharmvar.org/.
- Were only two CYP2D6 variants tested? It seems like that is the case in lines 126-127, but in table S1 you also show duplications. Please clarify. If the other alleles/variants are unlikely to occur in your population, that should be stated. This should also be discussed as a limitation.
- There is no mention of potential importance of induction for CYP1A2 and how that may influence results (or not influence results, although you do note that it plays a much smaller role in duloxetine metabolism on lines 258-259, I think it could be argued that it should be addressed briefly).
- Were patients on any other (non-psychotropic) medications that could have led to phenoconversion?
- A reference is missing on line 101.
Other thoughts: it seems like that first table should be in the text and not in the supplement (but it seems to be in both?). Does DMI predict UKU SERS relatively better than response?
Author Response
- In this manuscript, the authors provide an overview of their study associating duloxetine response with CYP2D6 and CYP1A2 metabolizing capacity (using calculated scores reflecting predicted enzyme activity), side effects, and a number of additional clinical factors such as somatic conditions and sex. They determined that higher CYP2D6 metabolism and less side effect burden were associated with better response.
- This is an interesting and timely paper given that pharmacogenetic panel tests being sold to guide psychotropic treatment sometimes include CYP1A2 and CYP2D6 when formulating recommendations for duloxetine use, based on limited evidence
- However, there are several issues that should be addressed before publication
- The data on number of CYP2D6 and CYP1A2 variants is outdated: https://www.pharmvar.org/.
Thank you for this crucial remark. We have updated data on the number of CYP2D6 and CYP1A2 and changed the reference for https://www.pharmvar.org/.
“So far, 172 variants of the CYP2D6 gene have been identified [15]. The CYP1A2 is involved in the metabolism of several clinically important drugs, including numerous antidepressants. To date, 21 various CYP1A2 alleles and corresponding metabolic subtypes have been described [16].”
- Were only two CYP2D6 variants tested? It seems like that is the case in lines 126-127, but in table S1 you also show duplications. Please clarify. If the other alleles/variants are unlikely to occur in your population, that should be stated. This should also be discussed as a limitation.
Thank you for this crucial remark. We have added additional information to the Methods section in order to clarify the number of tested polymorphisms:
“Alleles selected for further research were the most common functionally important alleles in the Caucasian population, selected on the basis of global databases, including PharmGKB [22] and CPIC guidelines [23] according to the state of knowledge current for 2017, when the study was planned. We decided to narrow the scope of our study to the two polymorphisms of the CYP2D6 gene and three in the CYP1A2 gene. For Caucasians, the CYP2D6 allele *3 has the mean frequency of 1.6% and the allele *4 of 18.5% whereas the CYP1A2 allele 1F has the mean frequency of 33% and the allele *3 of 1%. In turn, the frequency of the CYP1A2 allele 1D varies between 3.4 and 11% [24-26].”
Additionaly we added a following sentence in limitations subsection:
“In the described study, we did not take into account the CYP2D6 and CYP1A2 variants sporadically occurring in the Polish population, which should be perceived as a limitation.”
- There is no mention of potential importance of induction for CYP1A2 and how that may influence results (or not influence results, although you do note that it plays a much smaller role in duloxetine metabolism on lines 258-259, I think it could be argued that it should be addressed briefly).
Thank you for this valuable suggestion. We have added a paragraph related to induction of the CYP1A2:
“It is also important to consider the potential phenomenon of cytochrome P450 induction. Notably, CYP1A2 has been documented to demonstrate inducibility at both the mRNA and protein levels in response to a spectrum of chemical agents, including omeprazole, lansoprazole, 2,3,7,8-tetrachlorodibenzo-p-dioxin (TCDD), 3-methylcholanthrene, and rifampicin [37]. However, it should be noted that none of the subjects enrolled in this study were administered these specific drugs. Additionally, it is noteworthy that some reports suggest an augmentation of the CYP2D6 activity, typically considered non-inducible, following exposure to common valerian (in a linear dose-dependent manner) and Ginkgo biloba (with the highest concentration) extracts [38]. It is essential to highlight that herbal dietary supplements were not permitted within the study cohort. Furthermore, in the context of drug metabolism, it is relevant to acknowledge the influence of cigarette smoking, which can expedite the metabolism of certain drugs, particularly those predominantly metabolized by CYP1A2. Notwithstanding, as demonstrated in the study by Hukkanen et al. [39], it should be clarified that nicotine itself does not contribute to the induction of CYP1A2. In sum, the phenomenon of metabolic induction and phenoconversion likely had a negligible impact on the results of our study.”
- Were patients on any other (non-psychotropic) medications that could have led to phenoconversion?
Thank you for this valuable suggestion. We added detailed information about other medications in Table 1.
Table 1. General characteristics of the sample.
|
Variable |
Mean ± SD or n (%) |
|
Age |
41.6 ± 18.4 |
|
Sex (female) |
57 (57.0%) |
|
Marital status (married) |
37 (37.0%) |
|
Education (higher education) |
65 (65.0%) |
|
Somatic conditions |
30 (30.0%) |
|
Diabetes drugs |
5 (5.0%) |
|
Drugs for hypertension |
12 (12.0%) |
|
Asthma drugs |
6 (6.0%) |
|
Chronic obstructive pulmonary disease drugs |
4 (4.0%) |
|
Coronary artery disease drugs |
3 (3.0%) |
|
Rheumatoid arthritis drugs |
2 (2.0%) |
|
Drugs for chronic migraines |
8 (8.0%) |
|
Drugs for osteoarthritis |
5 (5.0%) |
|
Drugs for chronic back pain |
7 (7.0%) |
|
Other drugs |
5 (5.0%). |
|
UKU SERS |
5.8 ± 7.6 |
|
HAM-D (baseline score) |
21.6 ± 9.3 |
|
HAM-D (reduction, %) |
52.3 ± 40.9 |
|
HAM-A (baseline score) |
17.3 ± 7.5 |
|
HAM-A (reduction, %) |
51.1 ±37.4 |
Data expressed as n (%) or mean (SD).
We have also discussed that phenoconversion had a negligible impact on the results of our study:
“It is also important to consider the potential phenomenon of cytochrome P450 induction. Notably, CYP1A2 has been documented to demonstrate inducibility at both the mRNA and protein levels in response to a spectrum of chemical agents, including omeprazole, lansoprazole, 2,3,7,8-tetrachlorodibenzo-p-dioxin (TCDD), 3-methylcholanthrene, and rifampicin [37]. However, it should be noted that none of the subjects enrolled in this study were administered these specific drugs. (…) In sum, the phenomenon of metabolic induction and phenoconversion likely had a negligible impact on the results of our study.”
- A reference is missing on line 101.
Thank you for your vigilance. We have added needed references and revised the format of references.
Other thoughts: it seems like that first table should be in the text and not in the supplement (but it seems to be in both?).
Thank you for your vigilance, first table is going to be in the text, not in supplement.
Does DMI predict UKU SERS relatively better than response?
Thank you for this remark. In the conducted linear regression analysis, with the dependent variable being the total UKU-SERS score and analogous independent variables applied, no significant predictive impact of DMI for CYP1A2 (beta=0.069, p=0.500) and CYP2D6 (beta=-0.109, p=0.260) was observed - results are derived from the third model. In summary, according to our study, DMI appears to have greater predictive value in relation to the reduction of anxiety symptoms as compared to treatment tolerance during duloxetine therapy for depressive disorders.
Reviewer 3 Report
The present study investigated the relationships between genetic polymorphisms, clinical variables, and treatment outcomes. It makes significant contribution to the research area of this study. Thus, I recommend publication of this paper after minor revision. However, the paper needs further scientific editing to make it more professional.
Need some editings. Please read throughout the manuscript to make further editings, so that the quality of English language can be improved.
Author Response
The present study investigated the relationships between genetic polymorphisms, clinical variables, and treatment outcomes. It makes significant contribution to the research area of this study. Thus, I recommend publication of this paper after minor revision. However, the paper needs further scientific editing to make it more professional.
Comments on the Quality of English Language
Need some editings. Please read throughout the manuscript to make further editings, so that the quality of English language can be improved.
Thank you for your vigilance. We have performed additional language editing.
Round 2
Reviewer 1 Report
Authors have modified the manuscript according to reviewer's suggestions.
Author Response
.
Reviewer 2 Report
I appreciate the authors attention to detail in addressing comments from the reviewers. Overall, I think their additions have greatly strengthened the paper, but there is one area where I think it is important to add more information:
You summarized concerns related to CYP1A2 inducibility, and correctly point out that smoking, not just nicotine, can induce CYP1A2. Coud you please note if any of your patients were smoking?
Author Response
I appreciate the authors attention to detail in addressing comments from the reviewers. Overall, I think their additions have greatly strengthened the paper, but there is one area where I think it is important to add more information:
You summarized concerns related to CYP1A2 inducibility, and correctly point out that smoking, not just nicotine, can induce CYP1A2. Coud you please note if any of your patients were smoking?
Thank you for this crucial remark. We have added to Table 1 information about cigarette smoking (n=27, 27.0%).